# Obesity Risk Assessment for Spanish-Speaking Immigrant Families with Young Children in the United States: Reliability and Validity with Nutrient Values

**DOI:** 10.3390/children10050868

**Published:** 2023-05-12

**Authors:** Marilyn S. Townsend, Mical K. Shilts, Louise Lanoue, Christiana Drake, L. Karina Díaz Rios, Nancy L. Keim, Dennis M. Styne, Lenna L. Ontai

**Affiliations:** 1Department of Nutrition, University of California, Davis, CA 95616, USA; 2Department of Family and Consumer Sciences, Nutrition, Food & Dietetics Program, California State University, Sacramento, CA 95819, USA; shiltsm@csus.edu; 3Department of Statistics, University of California, Davis, CA 95616, USA; cmdrake@ucdavis.edu; 4Division of Agriculture and Natural Resources, Public Health Department, University of California, Merced, CA 95343, USA; kdiazrios@ucmerced.edu; 5USDA Western Human Nutrition and Research Center, University of California, Davis, CA 95616, USA; nancy.keim@usda.gov; 6Department of Pediatrics, University of California Medical Center, Davis, CA 95817, USA; 7Department of Human Ecology, University of California, Davis, CA 95616, USA

**Keywords:** overweight, obesity, young children, preschool, child obesity prevention, validation, evaluation, validity, reliability, low-income families, nutrition education, Hispanic, Latino

## Abstract

The purpose is to examine validity and reliability for an obesity risk assessment tool developed in Spanish for immigrant families with children, 3–5 years old using an 8-week cross-sectional design with data collected over 1 year at Head Start and Special Supplemental Nutrition Program for Women, Infants and Children [WIC]. Parent/child dyads (206) provided a child obesity risk assessment, three child modified 24 h dietary recalls, three child 36+ h activity logs and one parent food behavior checklist. Main outcome measures were convergent validity with nutrients, cup equivalents, and diet quality and three assessments of reliability that included item difficulty index, item discrimination index, and coefficient of variation. Validity was demonstrated for assessment tool, named *Niños Sanos*. Scales were significantly related to variables in direction hypothesized [*p* ≤ 0.05]: Healthy Eating Index, fruit/vegetable cup equivalents, folate, dairy cup equivalents, vitamins D, β-carotene, fiber, saturated fat, sugar, time at screen/ sleep/physical activity and parent behaviors. Three measures of reliability were acceptable. The addition of nutrient values as an analytical validation approach adds strength and consistency to previously reported *Niños Sanos* validation results using children’s blood biomarkers and body mass index. This tool can be used by health professionals as an assessment of obesity risk in several capacities: (1) screener for counseling in a clinic, (2) large survey, (3) guide for participant goal setting and tailoring interventions, and (4) evaluation.

## 1. Introduction

### 1.1. Background

Obesity prevalence (≥95th percentile-for-age) is significantly higher among Hispanic youth (23.6%) compared with non-Hispanic white youth (14.7%) in the United States [1], thus putting Hispanic youth at higher risk for obesity-associated medical conditions [2]. The differences in obesity rates appear early, as indicated by data among preschool children ages 3–5 years: 16.5% Hispanic vs. 9.9% non-Hispanic [1]. Some home environments are more conducive to setting young children on trajectories for unhealthful weight gain, overweight and eventually obesity [3,4]. The American Academy of Pediatrics (AAP) and the Institute of Medicine (IOM) recommend early intervention of modifiable environmental and behavioral factors associated with the risk of pediatric obesity [3,5]. Accordingly, valid assessments to screen for children at risk, evaluate behavioral family interventions and serve as a counseling tool for goal setting and tailoring intervention efforts are essential. Hispanic/Latinos represent 19% of the United States population and are the ethnic majority in the state of California (40.2%) [6]. Assessment tools that are tailored for and validated with Hispanic/Latino families with young children are scarce, yet necessary to sensibly address obesity prevention and nutrition and health inequities affecting a substantial proportion of the population.

### 1.2. Healthy Kids

Building on the IOM and AAP recommendations to address pediatric obesity in the U.S., our earlier research uncovered behaviors related to pediatric obesity occurring in the family environment through literature reviews [7]. This resulted in identifying questions in the domains of diet, lifestyle (i.e., sleep, screen time, physical activity), and parenting. The resulting assessment tool for low-income English-speaking parents with 3–5 year-old children, named *Healthy Kids* (HK), underwent reliability and validity testing using objective measures (inflammatory biomarkers [8] and measured body mass index [9]) and subjective measures (24 h dietary recalls [9] and 36 h activity logs [9]) in a longitudinal research design. Results showed that children with less healthful HK scores had an elevated inflammation index, indicating a low-grade chronic systemic inflammatory state. In children with less healthful behaviors, the inflammatory index increased over the 12 months of the study (*p* = 0.007), but not in children with more healthful profiles [8]. Results indicated an inverse relationship with a lower and healthier BMI percentile-for-age associated with a higher and healthier score (*p* = 0.02) [9].

### 1.3. Niños Sanos

Research from the above work was then rigorously revised to adapt it for another audience: Spanish-speaking immigrant families with young children, 3–5 years old, with a range of reading and written literacy skills in Spanish. Validation of a Spanish-language tool to target this population is warranted given that most low-income immigrants in the U.S. originate from Mexico and primarily speak Spanish, with about 40% having limited English proficiency and educational attainment levels less than a high school diploma [10]. The Mexican origin is especially pronounced in families with young children, which represent a majority of recipients for services targeting families with young children, such as Special Supplemental Nutrition Program for Women, Infants, and Children (WIC; 62%) [11] and Head Start (38%) [12]. The English version of *Healthy Kids* was converted to Spanish using forward translation conducted by 2 bilingual and bicultural Spanish speakers. To establish face validity of the converted tool, equivalence verification and cognitive interviews to elicit participant views on item clarity and understandability were employed to establish linguistic and cultural appropriateness of the text [13]. Details of this process are reported elsewhere [14]. The original photos targeting English speaking families were retaken with culturally appropriate images of families, activities and food items as recommended by the bilingual translators and endorsed by participants. Pairing the item text with a visual has been previously shown to improve reader understanding of self-administered questionnaires, thus reducing cognitive load and enhancing readability [15,16,17].

The resulting obesity risk assessment tool is named *Niños Sanos* (translation: Healthy Kids) and contains 18 items and potential minimum and maximum total scores of 18 and 90 points, respectively. Response options were coded on a scale of 1 to 5 points and recoded, if necessary, to reflect 5 points as the healthiest response option. Details of the selection process for the final 18 items are reported elsewhere [14].

Criterion validity of *Niños Sanos* was established using BMI percentile-for-age (*p* < 0.05) and blood biomarkers for metabolic (*p* = 0.03), lipid (*p* = 0.05), and anti-inflammatory health (*p* = 0.047) [14]. These results demonstrated that children with higher *Niños Sanos* scores had more desirable biomarker profiles and body composition. Details of these analyses are reported elsewhere [14]. The final version of *Niños Sanos* is available for researchers and practitioners at the link provided [18] and a photo is shown in Figure 1.

### 1.4. Purpose

Extending the above research [14], this current paper describes the work to establish convergent validity using dietary, activity and sleep data, and explore reliability using three tests. Specifically, objectives for validation are to establish: (1) the Niños Sanos dietary scale and sub-scale scores association with nutrient variables in the healthful direction; (2) the Niños Sanos dietary scale score association with the Healthy Eating Index total score; (3) the Niños Sanos dietary scale correspondence to parent food behaviors; and (4) the Niños Sanos physical activity, screen time and sleep scale scores agreement with variables calculated from the mean of three child’s 36+ h physical activity/screen time/bedtime logs. In addition, three reliability tests (i.e., item difficulty index, item discrimination index, and coefficient of variation) are examined to establish Niños Sanos consistency of measurement. Readability and respondent burden results are investigated.

## 2. Methods

### 2.1. Framework

The biopsychosocial framework used for the design of this study was based on Systems Theory [19,20]. This framework considered the health of the child in the context of the family environment and in that respect is similar to the Socio-Ecological Model (SEM) used to guide development of the tool’s content. Details are reported elsewhere [14]. The framework is provided as a Appendix A.

### 2.2. Participants

Participants (n = 273 parent-child dyads) lived in 2 urban counties in northern California. The adults were parents or caregivers, ≥18 years, who declared Spanish as their preferred language, had at least one child aged 3–5 years and participated in at least one federal assistance program for low-income residents.

### 2.3. Recruitment Plan 

Incorporating the concepts described by Diaz Rios and Chapman-Novakofski into the recruitment plan [21], parents were recruited by direct solicitation and at information sessions held at Head Start (n = 21) and WIC sites (n = 3). Three 2nd generation adults [22] ages 21–29, female, whose primary language spoken at home during childhood was Spanish and whose parents originated from Mexico, were employed and trained for parent/child recruitment and data collection. They were bicultural and bilingual, first in Spanish and second in English. To enhance trust during recruitment, staff met with potential participants at trusted locations where Spanish-speaking parents bring their children for Head Start classes or to pick up vouchers for WIC. Although signage about the study was posted at these sites and particulars were discussed at Head Start parent meetings by UC Davis research staff, recruitment was always conducted one-on-one in-person [21].

### 2.4. Retention Plan

Recruitment and subsequent data collection took place during 2017 and 2018 when stigma related to immigration status was a topic of conversation among Hispanic families [23]. Retention was a concern, so data collection staff acknowledged parent fears and conveyed confidentiality assurances. Participant time was respected by offering interview appointments at the time and place of participant preference. Trusted staff from Head Start interacted with parents about the study and made announcements at Head Start parent meetings to encourage parents to remain in the study. When passive channels of communication were employed, those channels were always held in combination with face-to-face contact [21].

### 2.5. Study Design, Timeline & Data Collection

Data for an individual parent/child dyad were collected over an 8-week period at 5 data collection points interspaced approximately 1–2 weeks apart as shown in Appendix A. At Week 1, parents completed a demographic questionnaire as well as the first child’s 36+ h physical activity/screen time/bedtime log, and child’s 24 h dietary recall [24]. The 36+ h activity log and 24 h diet recall were administered again at Week 5 and at Week 7. *Niños Sanos* [14,18] was completed by parents at Week 3 along with a parent food behavior checklist, *Lista de Habitos Alimenticios* (translation: Food Behavior Checklist), previously shown to be valid with Spanish-speaking immigrant adults [25,26]. 

The 5 data collection sessions lasted approximately one hour and were conducted in the language preferred by the parent. Data were collected in person and by telephone interview. Written documentation such as consent forms, assessment tools, questionnaires and logs were printed in Spanish.

### 2.6. Validation

#### 2.6.1. Convergent Validation for Dietary Behaviors with 24 h Dietary Recalls

Recalls were collected in Spanish by trained researchers in person and by telephone using the Automated Self-Administered 24 h Dietary Assessment Tool (ASA24 system) version 2014 employing USDA’s Automated Multiple-Pass Method (AMPM) and the Food and Nutrient Database for Dietary Studies (FNDDS) v4.1. Recalls were interviewer-administered, replacing the self-administered protocol outlined by ASA24. The ASA24 software protocol was used by the interviewer to guide the parent interview about the child’s diet. This deviation from standard protocol was necessary for several reasons: (1) Minimize data entry error related to participants’ proficiency with electronic devices for ASA24; (2) Parents do not read English; and FNDDS database lacked foods commonly consumed by Spanish-speaking immigrant families resulting in foods without matches. These foods were tagged for later discussion by researchers to determine the best substitute food item. To ensure data integrity, registered dietitians on the research team provided extensive training and guidance to the interviewers who collected the child’s dietary data from the parent in Spanish and then back-translated to English before entering the recall data into ASA24. Parents were asked to report child snack and meal items consumed in their presence. Foods provided by Head Start or another provider were not included when the parent was not present. Two weekday and one weekend recalls were collected per child at weeks 1, 5 and 7 of the 8-week protocol (Appendix A). The Healthy Eating Index (HEI) version 2015 was calculated using data reported on the 24 h recalls [27].

#### 2.6.2. Convergent Validation for Physical Activity, Screen and Sleep Behavior Items with Child 36+ h Logs

The child’s behaviors were assessed using the University of California *My Child’s Food and Activity Diary*, a 36+ h activity log [24]. Variables were collected or calculated for time of day, duration in minutes and location for activities that included bedtime, wake time, nap times, screen activities and physical activity. Two non-consecutive weekdays and one weekend day, during weeks 1, 5 and 7 respectively, were selected by the parent for data collection (Appendix A).

#### 2.6.3. Concurrent Validation with Parent Food Behaviors

The parent’s food-related behaviors were assessed using the 16-item University of California *Lista de Habitos Alimenticios*, a food behavior checklist for Spanish speakers, previously shown to be valid using 3 researcher-interview 24 h dietary recalls. Details are available [25,26]. Parent responses were coded using 4-response options per item for a maximum of 4 points per item. The 16 items were scaled and summed for a maximum of 64 points [26].

### 2.7. Reliability

Three approaches to reliability were assessed to give different perspectives on the consistency of parent responses to *Niños Sanos* [28,29,30].

(1) Item difficulty index [30], the ratio of each item’s mean to its maximum, was generated by dividing the mean value of responses to the item by the maximum score of 5 representing the most healthful response, for a potential range of 0.20–1.0.

(2) Item discrimination index [28] represents the ability of an item to discriminate between those parent/child dyads who do well on *Niños Sanos*, i.e., have high total scores, and those who do not. The index is the corrected item-total correlation, with a potential range of −1.0 to +1.0. The higher the value, the better the item is at identifying families with overall obesity prevention behaviors, indicated by the other 17 items. A value close to zero would indicate the item has no relationship with the other 17 obesity prevention items on the tool. A negative value would indicate the item is negatively related to the other 17 items; this item would be considered for removal.

(3) The coefficient of variation (CV) is the ratio of the standard deviation to the mean and is expressed as a percentage from 0–100% [31]. The higher the coefficient of variation, the greater the level of dispersion around the mean. The lower the value of the coefficient of variation, the more precision associated with the estimate, in this case, the item mean.

### 2.8. Readability and Respondent Burden

Additionally, two properties considered important for this audience and setting were assessed: readability [32] of *Ninos Sanos* by the Flesch-Kincaid Reading Index and Flesch Reading Ease using Microsoft Word software for Spanish (MS Office Professional Pro 2016, Microsoft, Inc., Seattle, WA, USA) and respondent burden by timing respondents completing the self-administered tool.

### 2.9. Statistics

Categorical demographic data are reported as number of participants and % of sample, using SAS for Windows, version 9.4 (SAS Institute, Cary NC, USA, 2017). Statistical program R [https//:cran.cnr.berkeley.edu, (accessed on 2 February 2020) V. 2014, Vienna, Austria] was used to calculate Pearson’s Chi-squared test with Yates’ continuity correction to compare demographic data for dropout and final samples. 

Parent responses to *Niños Sanos* were next scored and total scores were discretized into 2 groups of low and high total behavior scores (18 to 74.9 points; 75.0 to 90 points) for some analyses. For other analyses, continuous variables were used. Kruskal Wallis (KW), a one-factor analysis of variance test, was used to compare *Niños Sanos* scores with dietary parameters and then with screen, activity and sleep parameters. Following KW analyses, Spearman rank order correlation coefficients were calculated. Spearman rank correlation coefficients provided the direction and strength of relationship between two quantitative variables. Significance level was set at *p* ≤ 0.05 for all analyses.

Tests for reliability reported in were conducted with data using SPSS [version 26.0, SPSS Inc., Chicago, IL, USA, 2020].

The study was powered by an analysis with BMI percentile-for-age as part of the broader *Niños Sanos* research. Those results are reported elsewhere [14]. The dietary results reported in this report were secondary to the main hypothesis about BMI.

### 2.10. Ethical Conduct

The latest version of the protocol, ID #693978-15 dated 27 June 2020, was approved by the Institutional Review Board of the University of California at Davis until 25
June 2029. Participants provided a signed consent at enrollment and were given monetary stipends reflective of their time and effort.

## 3. Results

### 3.1. Sample, Retention and Attrition

Families (n = 273) were enrolled in the study with 206 parent/child dyads completing all data collection time points with a 78% retention rate. Tests for attrition (*p* ≤ 0.05) revealed that participants who remained in the study over the 8-week time period were not different for most demographic variables tested with two exceptions. These parents were older by +2 years and more likely to report being unemployed compared to parents who dropped from the study (n = 67). Additional tests showed no difference in *Niños Sanos* scores between parents remaining in the study (n = 241) and those (n = 32) that dropped after week 1 (*p* = 0.63) (Appendix A). From the final data set, 4 parents did not complete one or more of the 18 items on *Niños Sanos*.

### 3.2. Demographics including Generational Status

Parents (n = 206) were predominantly low-income and Spanish-speaking as a first language. They self-identified as speaking Spanish (87%), English (6%) or both Spanish & English (7%) in the home. Children (54% female) were born in the US (95%), averaged 45 months at enrollment. For acculturation variables, all parents were born outside the U.S. in Mexico (81%) and other Central American countries (19%). Parents (100%) were 1st generation based on IOM definitions. The majority of children were born in the U.S. (n = 198, 95%) and considered 2nd generation. Additional demographic characteristics are provided in Table 1.

### 3.3. Convergent Validation for Dietary Behaviors with 24 h Dietary Recalls

Fifteen items related to eating behavior composed the *Niños Sanos* dietary sub-scale (Appendix A). The hypothesized associations offering support for the validity of the dietary sub-scale included HEI total score (r = 0.33, *p* ≤ 0.0001) and HEI sub-scale for proteins (r = 0.23, *p* ≤ 0.003) among others with items grouped by corresponding nutrients and food group (Table 2). 

With regards to the 4-item fruit and vegetable sub-scale, key variables were significant: vegetable cup equivalents (r = 0.22, *p* = 0.03), fruit cup equivalents (r = 0.23, *p* ≤ 0.02), and total fiber (r = 0.21, *p* = 0.03). β-carotene was not (r = 0.19, *p* = NS). The dry bean item composing the bean sub-scale (which is also included as a vegetable item), was related to fiber (r = 0.25, *p* ≤ 0.002) and the HEI for green vegetables and beans (r = 0.42, *p* ≤ 0.0001). The milk sub-scale composed of two milk items was related to cup equivalents of milk (r = 0.19, *p* = 0.01). The 2-item sub-scale about soda, juice drinks and sports drinks was related to the HEI for added sugar (r = 0.19, *p =* 0.02) and inversely related to total dietary added sugar (r = −0.20, *p* = 0.008). The 3-item saturated fat sub-scale was weakly related to the HEI for saturated fat (r = 0.14, *p* ≤ 0.10). All associations between the categories and sub-scales from *Niños Sanos* and HEI scores are shown in Table 2.

### 3.4. Convergent Validation for Screen, Sleep, Physical Activity Behaviors with Child 36+ h Logs

The single *Niños Sanos* screen time variable represented by the television (TV) item was related to the screen time in minutes reported on the mean of three 36+ h logs (r = −0.44, *p* ≤ 0.0001) as shown in Table 2. The more screen time in minutes, the lower the score on the relevant NS item. The single *Niños Sanos* sleep time variable that captured children’s bedtime was related to average sleep time reported in the logs (r = −0.76, *p* = 0.008). The physical activity item was related to the mean activity time measured in minutes (r = 0.19, *p* = 0.05). 

### 3.5. Concurrent Validation with Parent Food Behaviors

As hypothesized, the parent food-related behaviors assessed by the 16 items on the University of California’s *Lista de Habitos Alimenticios* were significantly related to the child’s overall score on *Niños Sanos* (r = 0.57, *p* ≤ 0.0001). 

### 3.6. Reliability

Item difficulty index scores varied from 0.47 to 0.95. We considered the most desirable scores to be between ≥0.40 and ≤0.80. Ten items had scores within this range, while eight had scores > 0.80. One interpretation is that some behaviors, such as ‘Trimming fat from meat’ which was 0.86, were more routinely implemented behaviors and thus were relatively easier items to answer in comparison to other behaviors that may be less routine, such as ‘Parent drinks milk’. Additionally, social desirability bias in parent reporting is always a consideration. 

Item discrimination index was lowest (−0.021) for ‘My child drinks milk’ and highest (0.49) for ‘I buy vegetables’. The high score was because parents who indicated they buy vegetables also responded similarly to other *Niños Sanos* items. Conversely, parents are not responding as expected to ‘My child drinks milk.’ When parents reported healthful behaviors for their children, they didn’t often report milk consumption. The interpretation is that frequency of milk drinking is not related to other healthful behaviors such as eating vegetables. Parents did not respond as expected to “My child eats snack foods like cookies, chips and candy.” A low score of +0.12 indicated that parents who reported their children regularly consumed cookies, chips and candy also reported some healthful behaviors. With a score near zero, our interpretation is that consumption of cookies, chips and candy was not related to the presence of healthful or unhealthful behaviors. Overall, the item-discrimination results showed that there was wide variation in the practice of behaviors related to obesity prevention across respondents, and a parent practicing one healthy behavior did not necessarily practice other healthy behaviors. Results demonstrate that preferences for protective obesity prevention behaviors vary by family (Table 3).

Item coefficient of variation (CV) scores indicate dispersion around the mean for responses to a given item, with smaller scores indicating more consistent measurement. The CV scores for *Niños Sanos* items ranged from 9% for “My child is not drinking soda” and “My child does not eat fast food” to 43% for “My child eats beans”. The two fast food items rendered consistent responses from parents compared to other NS items. The bean item was answered with the greatest variation of responses by participants (Table 3).

### 3.7. Readability and Other Psychometric Properties 

With the objective of producing a readable tool in Spanish for English-as-a-second-language (ESL) audiences, the photographic additions conveying the content of each item to the user allowing the number of words to be reduced improved the readability index score. *Flesch-Kincaid Reading Index for Spanish* readability score was 81 of a possible 100 points, and a *Flesh Reading Ease for Spanish* was a score of 80. Scores between 80–89 points indicate easy reading. The average number of syllables per word for *Niños Sanos* was 1.9 while average sentence length was under 5 words. The 18-item tool with 1- and 2-syllable words and pictorial format required 10–14 min to administer meeting our criteria for minimal respondent burden. 

## 4. Discussion

### 4.1. Validity

This current research provides support for the validation of *Niños Sanos,* an obesity risk assessment using a self-administered questionnaire format for immigrant Spanish speakers including those with limited Spanish literacy. The dietary and activity data support previously reported results that used the child’s body mass index and metabolic, inflammatory and lipid biomarkers [14]. The current results support the hypothesized associations between validation variables and *Niños Sanos* responses. A comprehensive picture of strength in support of a valid tool is portrayed using multiple sources of data and types of analysis.

Associations between the *Niños Sanos* total diet sub-scale score with an individual nutrient was generally moderate (r = 0.33, *p* = < 0.0001, HEI-Total Score) or adequate (r = 0.22, *p* = 0.002, fiber) from a statistical perspective (Table 2). However, when an individual nutrient is considered in combination with other hypothesized nutrients and cup equivalents, stronger associations with *Niños Sanos* scale scores emerge. There are several possible factors to explain these correlation coefficients. There is more than one food group source for most nutrients. Using fiber as an example, a high concordance cannot be expected for the relationship between *Niños Sanos* fruit and vegetable sub-scores and total dietary fiber, because fiber sources include whole grains, legumes, nuts and seeds, in addition to fruit and vegetables. 

### 4.2. Reliability

This current research also provides evidence about the quality of *Niños Sanos* with assessments of reliability. Psychometric tests of reliability can reveal relevant characteristics of individual items in addition to the consistency of responses across the total tool. For example, the data for the vegetable availability item are: mean of 4.2 ± 1.0 SD, 0.85 item difficulty, 0.49 item discrimination and 23% CV. The relatively high score for item difficulty indicates that parents report practicing this behavior with ease compared to lower scoring items. The result for item discrimination reveals that this item is the best choice among the 18 items to identity parents and children who practice healthful behaviors overall. By comparison, the data for the vegetable accessibility item was: mean of 3.2 ± 1.3 SD, 0.63 item difficulty, 0.45 item discrimination and 41% CV. The item difficulty score indicates this behavior is practiced with less frequency. Both items discriminate in a similar manner meaning the items equally identify parents with higher total scores indicating more healthful behaviors reported. This item has a smaller mean and larger standard deviation compared to vegetable availability. By comparison, the vegetable accessibility item has a larger %CV. Our interpretation is that the vegetable accessibility item generates more variation among parent responses in this sample compared to the vegetable availability item. If a shorter tool is desired, vegetable accessibility is the preferred item for risk assessment.

Response option formatting has been a topic of cross-cultural psychological research with this Spanish-speaking audience. Researchers reported that Latinos, including Mexican Americans, with low levels of education and acculturation tended to select the extreme response set, i.e., the highest and lowest ends of the response scale, and to demonstrate acquiescence, i.e., choosing the ‘yes’ response [13,33,34,35]. The results from our current work indicate that most of our participants responded with the 3rd and 4th response options and not the extremes, 1st and 5th. And compared to our previous study assessing obesity risk behaviors with an English speaking audience, means for the item responses for 7 of the 8 overlapping items were smaller with Spanish-speaking participants [36]. This reinforces our observation that participants did not over-select extreme response options in our study. At the same time, participants tended to demonstrate acquiescence, agreement with more positive options.

### 4.3. Risk Assessment vs. Evaluation

The psychometric procedures employed for evaluating an item’s effectiveness depends on the purpose of the assessment and the researcher’s preference. To assess obesity risk, the tool would be administered to parent/child dyads on one occasion. The ideal item for such a tool has the following characteristics: valid using externally determined variables such as BMI and blood biomarker values as well as valid using self-report analysis such as 36 h logs and 24 h dietary recalls; reliable with sufficient variability; reliable with items that discriminate. If a practitioner could only ask one item to identify at risk families, the single item should have a relatively high score on the item discrimination index such as cooking from scratch and fruit availability in the case of NS. Developers of risk assessment using this pool of items would find items with relatively larger CV’s more helpful, such as the whole grains and vegetable/beans items. 

For the purpose of program evaluation for an intervention, the tool would be administered twice: once before and once after an obesity prevention education intervention. Item difficulty is important with relatively lower scoring items preferred in addition to the validations mentioned above. Items with both lower item difficulty scores and smaller % CVs, such as the sleep/bedtime, would improve the chance of the practitioner capturing change and documenting positive impact following the intervention. 

### 4.4. Comparison of Niños Sanos to Other Research

The current results were compared to those reported by the same researchers for the English-language *Healthy Kids*. Note that the previously reported validation research for both tools began with the same 43 items identified from the literature. Both tools have been shown to be valid using nutrient values [9], BMI [9,14] and blood values [8,14]. Overall, results were similar; however, details varied by audience. There are differences between the final versions of the two tools. The final *Niños Sanos* tool contains 18 items [18]; while *Healthy Kids* has 19 items. Eight of the *Niños Sanos* items do not appear on *Healthy Kids.*

Two tools for children developed by other researchers were located in the literature for obesity prevention in the United States. The well-researched 21-item Family Nutrition and Physical Activity (FNPA) tool in English targeted slightly older children, 6–12 years-old. This tool was validated with children (n = 704) using measured BMI over 6-year period in a mixed-income school district [37]. Children (n = 415) attending low-income schools were assessed for measured BMI, percent body fat and Acanthosis Nigricans [38]. The tool was later assessed with English-speaking parents of 2–9 year olds in a medical clinic setting and found to be effective with 2–5 year olds [39]. The second tool is the 15-item evaluation tool in English for the Expanded Food and Nutrition Education Program (EFNEP), which targets children 3–11 years-old, and was validated with low-income families (n = 62) [40,41]. Both of these tools were translated into Spanish and in the case of Dickin et al., with cognitive testing. Of these tools, only *Niños Sanos* was translated with cognitive testing and then in a separate study, validated with a Spanish-speaking immigrant sample [14].Tools have been developed in other counties for local populations: 4–16 year olds in Australia [42]; 4–11 year olds in England [43]; and 1–3 year olds in Ireland [44]. In their literature review, Bell, Golley and Magarey reported on short tools for dietary intake in Norway, Ghana, Europe, Switzerland, Germany, Belgium, Canada, and Argentina [45].

### 4.5. Limitations and Strengths

As with all studies, this one has limitations. First, generalizability of results beyond this convenience sample is limited; our sample was not randomly drawn from all low-income Spanish-speaking immigrant parents and their children in California. Thus, external validity of these findings to other Spanish-speaking audiences with limited resources is unknown. Second, the diet recalls have the inevitable measurement error associated with self-report. We sought to minimize this measurement error by employing bilingual and bicultural data collection staff and providing them with extensive training. Third, parents volunteered to participate in the study, thus the variability of parent responses in this cohort may be reduced compared to that of the general Spanish-speaking target population in California. Consequently, selection bias must be considered as a threat to external validity [46]. Fourth, the presence of social desirability bias may impact internal validity [47]. This source of error would suggest over-reporting of healthy behaviors such as vegetable accessibility and fruit availability and underreporting of unhealthy behaviors such as snacking on foods like cookies, chips and candy. When examining the potential impact of social desirability bias on *Niños Sanos*, this error would have the effect of reducing the level of statistical significance of results [46,47]. Fifth, there is the potential for the Hawthorne Effect to influence parent responses to the 24 h diet recalls and the 36+ h activity and sleep logs and thus impact internal validity [46]. Finally, the research design was cross-sectional; thus causality cannot be inferred [46]. Despite these limitations, important strengths should be noted. First, diet quality was characterized by the Healthy Eating Index, a validated diet quality indicator that is recognized by health professionals [48]. Second, by assessing overall diet quality, the effect of all nutrients grouped into one index is provided. And indices are more strongly related to mortality than any one component of diet alone [49]. Third, multiple approaches to validity and to reliability were employed, providing a more robust picture of the accuracy and consistency of *Niños Sanos* responses.

## 5. Conclusions

*Niños Sanos,* ready for field use by researchers and health practitioners, has demonstrated utility for obesity risk assessment among Spanish-speaking immigrant families with young children and is unique in that regard. These new results support and extend the previously reported validation with children’s BMI and blood biomarkers for obesity [14]. The addition of nutrient values as an analytical approach adds strength and consistency to the previous findings. *Niños Sanos* can be used by health professionals as an assessment of obesity risk in several capacities: (1) screener for counseling in health clinic settings for children most at risk [50], (2) large survey, (3) guide for participant goal setting and tailoring intervention efforts [51] and (4) evaluation tool to assess intervention impact. 

## Figures and Tables

**Figure 1 children-10-00868-f001:**
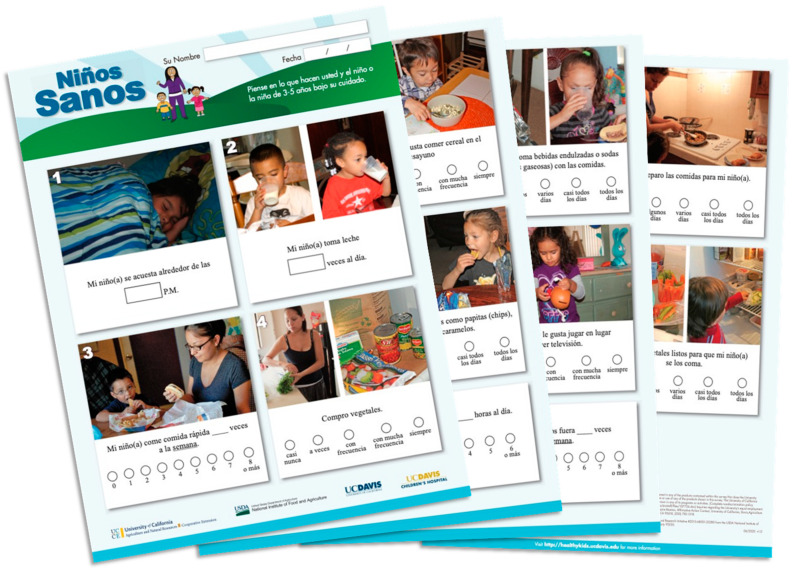
Niños Sanos Obesity Risk Assessment, final version with 18 items, tailored to Spanish-speaking immigrant parents of 3–5 year old children. Items include parent fruit availability, child fruit intake, parent vegetable availability & accessibility, child eating dry cooked beans, child milk intake, child whole grain intake, child sugar-sweetened beverage frequency, child fast food frequency, child fast food frequency, parent trimming fat, child energy dense snacking, child sedentary time, among others. Printed in color on one 11 by 17-inch paper, folded in booklet format.

**Table 1 children-10-00868-t001:** Descriptive statistics of *Niños Sanos* study participants.

Parent/Guardian ^a,b^		Child ^a,b^		Household ^a,b^	
**Gender** (Female)	206 (100)	**Gender** (Female)	113 (54.3)		
**Age** (years)	33.6 ± 6.0	**Age** (months)	51.8 ± 8.0		
**Marital status**					
Married	136 (65.4)				
**Ethnicity**		**Ethnicity**			
Hispanic/Latino	100	Hispanic/Latino	98.6		
**Socioeconomic Status ^a,b^**					
**Education**				**Household Income (monthly)**	
College degree	13 (6.4)			<$2000	118 (56.7)
Some college	30 (14.7)			$3000–3500	13 (6.3)
High school diploma	59 (28.9)			$3500–4000	5 (2.4)
**Employment**				>$4000	5 (2.4)
Unemployed	148 (71.1)			**Assistance programs** ^c^	
Seasonal	35 (16.8)			Head Start	166 (79.8)
Full time	25 (12.0)			WIC	172 (82.7)
				SNAP	84 (40.4)
				TANF	18 (8.6)
				NSLP	62 (29.8)
				Head Start	166 (79.8)
				WIC	172 (82.7)
**Health Characteristics ^a,b^**					
**BMI**		**BMI-for-age-percentile**			
Normal < 25	40 (19.5)	Underweight < 5th%ile	8 (4.0)		
Overweight (25–30)	75 (36.6)	Normal weight < 85th%ile	130 (65.0)		
Obese (30–40)	77 (37.6)	Overweight > 85th%ile	24 (12.0)		
Morbidly obese > 40	13 (6.3)	Obese > 95th%ile	38 (19.0)		
**Acculturation Status ^a,b^**					
**Generation ^d^**		**Generation ^d^**		**Language spoken at home**	
1st non-English speaking	206 (99.9)	1st	0	English	13 (6.3)
1st English speaking	0	2nd non-English speaking	179 (86.1)	Spanish	179 (86.1)
2nd English speaking	NA	2nd English speaking	29 (14.0)	English or Spanish	16 (7.7)
**Country of birth**		**Country of birth**			
U.S.	0 (0)	U.S.	198 (95.2)		
Mexico	168 (80.1)	Mexico	7 (3.4)		
Other	40 (19.2)	Other	3 (1.4)		

^a.^ Categorical data are reported as number of participants (% of sample). ^b.^ Continuous data are reported as means (±standard deviations). ^c.^ The major requirement for federal program participation is income using a formula factoring in household size and income. ^d.^ Generational status was coded: 1st generation is an immigrant to the U.S. not preceded by parents or other family members and 2nd generation is the child born in the US of a 1st generation immigrant. Note: Parent and household variables are not shaded. Child variables are shaded blue. Abbreviations: TANF, Temporary Assistance for Needy Families; NSLP, National School Lunch Program; WIC, Supplemental Nutrition Assistance for Women, Infants and Children; SNAP, Supplemental Nutrition Assistance Program; NA, not applicable.

**Table 2 children-10-00868-t002:** Validation using relationship of *Niños Sanos* sub-scales/categories with hypothesized variables from the mean of 3 24 h diet recalls, 3 36+ h activity and sleep logs, and University of California Food Behavior Checklist for low-income Mexican immigrant mother/child pairs (n = 206).

Categories/Sub-Scales from 18-Item Niños Sanos,	Convergent Validity	Kruskal–Wallis ^a,b^	Spearman ^a,b^	
n (Items)		*p*-Value	*r*	*p*-Value
**TOTAL DIET ^c^, 15**				
	**FOOD ITEMS**			
	Cup equivalents, Vegetable	0.021	0.21	0.002
	Cup equivalents, Fruit	0.003	0.21	0.003
	Cup equivalents, Grains	0.050	—	NS
	HEI-2015 Whole Grains	NS	0.14	0.040
	HEI-2015 Total Proteins	0.003	0.23	0.0007
	HEI-2015 Total Score	0.0001	0.33	0.0001
	**NUTRIENTS**			
	Protein (g)	0.032	—	NS
	Fiber (g)	0.001	0.22	0.002
	**PARENT**			
	UC FBC ^d^	0.0001	0.57	0.00001
**FRUIT/VEGETABLE, 4**				
Vegetable availability+	**FOOD ITEMS**			
vegetable accessibility+	Cup equivalents, Vegetable	0.029	0.22	0.002
fruit intake+	Cup equivalents, Fruit	0.016	0.23	0.0008
fruit availability	HEI-2015 Total Vegetables	0.068	0.19	0.007
	HEI-2015 Total Fruits	0.020	0.20	0.004
	**NUTRIENTS**			
	ß-carotene (mcg)	NS	0.19	0.005
	Fiber (g)	0.033	0.21	0.003
**BEANS/VEGETABLE, 1**				
Dry cooked bean intake	**FOOD ITEMS**			
	HEI-2015 Green and Beans	0.0001	0.42	0.0001
	HEI-2015 Total Proteins	0.021	0.14	0.043
	**NUTRIENTS**			
	Fiber (g)	0.002	0.25	0.0003
**MILK, 2**				
Child milk frequency+	**FOOD ITEMS**			
parent milk frequency	Cup equivalents, Dairy	0.076	—	NS
	Cup equivalents, milk	0.012	0.19	0.0058
	HEI-2015 Total Dairy	NS	0.12	0.0735
**WHOLEGRAIN, 1**				
Milk with cereal	**FOOD ITEMS**			
	Whole Grain (g)	0.021	0.20	0.004
	HEI-2015 Refined Grains	NS	—	NS
	HEI-2015 Whole Grains	0.007	0.22	0.002
**SUGAR-SWEETENED BEVERAGES, 2**				
Soda frequency + sports	**FOOD ITEMS**			
drinks, punch frequency	HEI-2015 Added sugar	0.022	0.19	0.0059
	**NUTRIENTS**			
	Added sugar (g)	0.008	−0.20	0.0038
**FAT/SATURATED FAT, 3**				
Energy density + energy	**FOOD ITEMS**			
density + saturated fat	HEI-2015 Saturated Fat	0.097	0.14	0.0420
**SNACKS, 1**				
Energy dense snacks	**FOOD ITEMS**			
	HEI-2015 Saturated Fat	0.020	0.20	0.004
	**NUTRIENTS**			
	Saturated fat (g)	0.023	−0.19	0.0074
**SCREEN TIME, 1**				
Television	Average total TV (minutes)	0.0001	−0.44	0.0001
	FOOD ITEMS			
	HEI-2015 Green & Beans			
	HEI-2015 Total Proteins			
	NUTRIENTS			
	Fiber (g)			
**PHYSICAL ACTIVITY, 1**				
Play, sedentary time ^b^	Child physical activity, average of 3 days (minutes)	0.054	0.19	0.0060
**SLEEP ^b^, 1**				
Bedtime	Average total sleep (minutes)	0.0076	−0.76	0.0001

^a^ Kruskal–Wallis and Spearman rank order correlation estimated using three modified 24 h dietary recalls showing parent report of child’s diet. Results expressed per 1000 kcal. ^b^ Kruskal–Wallis and Spearman rank correlation estimated using 3 days of 36 h logs showing parent report of television time and computer time. Average sleep from logs with six bedtimes and six wakeup times. Average physical activity from 3 logs. ^c^ Of the 18 items in the final version of Niños Sanos, 15 are specific to dietary behaviors. ^d^ 16 items on the University of California Food Behavior Checklist include parent vegetable quantity, as snacks, at main meal and types; fruit quantity, types, as citrus; perception of diet quality; food insecurity, milk intake, on cereal; sugar sweetened beverages; fat on chicken and meat; fish; and food labels.

**Table 3 children-10-00868-t003:** Reliability results for 18-item Niños Sanos: Item and scale difficulty index, item discrimination index, item and scale coefficient of variation and scale internal consistency (n = 202).

Obesity-Related Behavioral Construct	Item Content	Item Difficulty Index ^‡^	Item Discrimination Index ^¥^	Item Coefficient of Variation ^†^	Internal Consistency ^€^
Range ^£^		0.20 to 1.0	−1.0 to +1.0	0 to 100%	0 to 1.0 alpha
Vegetable availability	Parent buys vegetables	0.85	0.49	23	…
Vegetable accessibility	Vegetables ready for child	0.63	0.45	41	…
Fruit intake	Parent eats fruit	0.65	0.32	28	…
Fruit availability	Parent buys fruit	0.89	0.50	19	
Vegetables/dry beans	Child eats beans	0.49	0.25	43	
Milk frequency	Child drinks milk	0.66	−0.021	25	
Milk frequency	Parent drinks milk	0.47	0.31	35	
Wholegrains	Child eats cereal	0.61	0.26	39	
Soda frequency	Child drinks soda	0.95	0.12	09	
Sports drinks, punch frequency	Child drinks sugared beverages	0.91	0.22	14	
Saturated fat/energy density	Child eats fast food	0.87	0.14	09	
Saturated fat/energy density	Trim fat	0.86	0.40	26	
Saturated fat/energy density Or Snacks	Snack foods	0.80	0.12	16	
Saturated fat/energy density	Eating out	0.83	0.13	20	
Parenting/vegetables, HEI	Parent cooks from scratch	0.93	0.44	16	
Sleep	Bedtime	0.59	0.23	26	
Screen time	Child watches television	0.74	0.035	17	
Physical activity	Child plays instead of TV	0.67	0.31	33	
Obesity risk scale 18 items		0.74 *	NA	9 **	α = 0.68

^‡^ Item Difficulty Index scores represent the mean value of the responses to the item divided by the maximum score of 5 representing the most healthful response, for a potential range of 0.20–1.0. n = 202. ^¥^ Item Discrimination Index is the corrected item-total correlation [item score ÷total scale score without item] for a potential range of −1.0 to +1.0. n = 202. ^†^ Item coefficient of variation is the item variation among participants expressed as a percentage from 0–100% and is the standard deviation divided by item mean multiplied by 100. n = 202. ^€^ Cronbach’s alpha is proportion of the variance attributable to the true score. ^£^ Possible minimum observable, possible maximum observable. * Scale difficulty index score represents the mean value of the total scores divided by 18 and multiplied by 100 for a potential range of 0–100%. ** Scale coefficient of variation is the scale variation among participants expressed as a percentage from 0–100% and is the standard deviation for the 18-item tool divided by the scale mean multiplied by 100. n = 202.

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
