# Peer review of "Obesity Risk Assessment for Spanish-Speaking Immigrant Families with Young Children in the United States: Reliability and Validity with Nutrient Values"

_children, 2023, doi:10.3390/children10050868_

Round 1
Reviewer 1 Report
I would like to express my gratitude for the opportunity to review the original article entitled "Obesity risk assessment for Spanish-speaking immigrant families with young children" (children-2357045). This article is submitted to the "Global and Public Health" section of the Special Issue "Childhood and Adolescent Obesity and Weight Management-Volume 3" in the journal Children.
This study addresses the important public health issue of childhood obesity.
The aim of this study was to examine the validity and reliability of an obesity risk assessment tool developed in Spanish for immigrant families with children aged 3-5 years old. Although the abstract presents this objective, the authors later present a different objective in the end of the introduction: that valid assessments to screen for children at risk, evaluate behavioral family interventions, and serve as a counseling tool for goal setting and tailoring intervention efforts are essential. I suggest that the objectives should be unified, and the one presented at the end of the introduction seems to be more aligned with the content of the study. In section 1.4, the objectives of the study are presented clearly.
In the methodology, the authors used objective measures such as inflammatory biomarkers and measured body mass index, as well as subjective measures such as 24-hour dietary recalls and 36-hour activity logs, using a longitudinal research design.
Overall, this is a well-planned and well-executed study with results that increase our understanding of the topic and have great applicability for reducing childhood obesity among Spanish-speaking immigrant families in the United States.
Author Response
Obesity risk assessment for Spanish speaking immigrant families with young children in the United States: reliability and validity with nutrient values
Author response to Reviewer 1 |
|
1. I would like to express my gratitude for the opportunity to review the original article entitled "Obesity risk assessment for Spanish-speaking immigrant families with young children" (children-2357045). This article is submitted to the "Global and Public Health" section of the Special Issue "Childhood and Adolescent Obesity and Weight Management-Volume 3" in the journal Children. This study addresses the important public health issue of childhood obesity. |
Thank you for reviewing our paper
|
2. The aim of this study was to examine the validity and reliability of an obesity risk assessment tool developed in Spanish for immigrant families with children aged 3-5 years old. Although the abstract presents this objective, the authors later present a different objective in the end of the introduction: that valid assessments to screen for children at risk, evaluate behavioral family interventions, and serve as a counseling tool for goal setting and tailoring intervention efforts are essential. I suggest that the objectives should be unified, and the one presented at the end of the introduction seems to be more aligned with the content of the study. In section 1.4, the objectives of the study are presented clearly. |
Regarding your note about unifying the objective in the abstract with the text at the end of the introduction. Because the specific objectives are lengthy in the text marked 1.4, adding that level of detail to the abstract would be awkward. In the abstract, our goal was to provide a general purpose for the study and leave the specific objectives to the paper’s background information. To that end, we modified the abstract by replacing specific ‘objective’ with the term general ‘purpose’. Hopefully, this word change provides clarity for the reader. We also added the term "convergent" to specify the type of validity tested and the three reliability tests conducted. |
3. In the methodology, the authors used objective measures such as inflammatory biomarkers and measured body mass index, as well as subjective measures such as 24-hour dietary recalls and 36-hour activity logs, using a longitudinal research design. Overall, this is a well-planned and well-executed study with results that increase our understanding of the topic and have great applicability for reducing childhood obesity among Spanish-speaking immigrant families in the United States. |
Thank you for the positive comments. |

Reviewer 2 Report
This study is testing the validity and reliability of an obesity risk assessment tool developed in Spanish for immigrant families with children ages 3-5. We comment as follows.
Overall, the English seems difficult to read. Please proofread the English text.
Abstract: Unstructured and too long. It is unclear what "three measures" refers to.
Objective "Ninos Sanos" should be mentioned in the Methods section.
The text is not well presented, for example, italicized, font size is different, etc. Please correct. Please correct.
Purpose: (3) comes after (1), (2), and (3). Does this mean there are 4 items in total?
Please indicate the ethical considerations collectively at the end of the Methods section.
The validation is listed in the Methods section, but I think it should be listed as a sub-item (it is at the same level as the research design).
As for the Index of Reliability, it would be easier to understand if it is presented as an equation.
I think there is no need to divide Discussion into s sections. I think that the validity of the assessment tool should be evaluated and discussed by assessment item (for each behavioral pattern), rather than grouped by assessment criterion (reliability criterion).
As a scientific paper, the English in this document is difficult to read.
Author Response
Author response to Reviewer 2 |
|
1. Overall, the English seems difficult to read. Please proofread the English text. |
The paper was reviewed by two separate native English-speaking colleagues and proofread using Grammarly. Changes were made throughout the manuscript for clarity and reading ease. |
2. Abstract: Unstructured and too long. |
We removed the header so that the abstract is now unstructured to meet the journal guidelines. To address length, we deleted 13 words from the last section of the abstract. |
2a. It is unclear what “three measures’ refers to. |
The details of the three reliability measures were added to the abstract. |
3. Objective "Ninos Sanos" should be mentioned in the Methods section. |
We have added more detail of the Ninos Sanos objective in the methods section. |
4. The text is not well presented, for example, italicized, font size is different, etc. Please correct. |
Completed |
5. Purpose: (3) comes after (1), (2), and (3). Does this mean there are 4 items in total? |
That was an author error. Thank you for catching it. The text now reads: Specifically, objectives for validation are: (1) the Niños Sanos dietary scale and subscale scores are associated with nutrient variables in the healthful direction; 2) the Niños Sano dietary scale score is positively associated with the Healthy Eating Index total score; 3) the Niños Sanos dietary scale is positively related to parent food behaviors; and (4) the Niños Sanos physical activity, screen time and sleep scale scores are positively associated with variables calculated from the mean of three child’s 36+ hour physical activity/screen time/bedtime logs. |
6. Please indicate the ethical considerations collectively at the end of the Methods section. |
We have revised the existing text in 2.4 methods/study design (lines 308-11) to now read: 2.10. Ethical conduct The protocol, ID #693978-15 dated June 27, 2020, was approved by the Institutional Review Board of the University of California at Davis until June 25, 2029 and participants provided a signed consent at enrollment. They were given monetary stipends reflective of their time and effort. We moved the text to the end of methods. We made a new section 2.12 Ethical conduct instead of placing it at the end of Methods in 2.11 statistics. |
7. The validation is listed in the Methods section, but I think it should be listed as a sub-item (it is at the same level as the research design). |
Completed. 2.6. Validation with 3 sub items. |
8. As for the Index of Reliability, it would be easier to understand if it is presented as an equation.
|
We provided a definition of each index in methods section 2.7, but did not repeat it in the Results (see lines #260-277). 1) Item difficulty index27, the ratio of each item’s mean to its maximum, was generated by dividing the mean value of responses to the item by the maximum score of 5 representing the most healthful response, for a potential range of 0.20-1.0. 2) Item discrimination index28 ………is the corrected item-total correlation, with a potential range of -1.0 to +1.0. …….. 3) The coefficient of variation (CV) is the ratio of the standard deviation to the mean and is expressed as a percentage from 0-100%. 30
|
9. I think there is no need to divide Discussion into s sections. I think that the validity of the assessment tool should be evaluated and discussed by assessment item (for each behavioral pattern), rather than grouped by assessment criterion (reliability criterion).
|
The discussion section has 9 paragraphs covering distinct topics. Using headings assists with clarity and organization and corresponds to headings in other sections of the manuscript. We prioritized discussion of the most relevant findings and their implications for research and practice. The authors believe it would be repetitive to present results and discussion by the 18-items and not the aim of the study. We justify this approach in section 4.1. |

Reviewer 3 Report
Authors have attempted to examine the validity and reliability for an obesity risk assessment tool for Spanish immigrant families with young children. There are several issues to be addressed before considering for publication.
First, the novelty of the present study should be elaborated. While HK has already been validated for English-speaking population, how many more beneficiaries will we have after validating HK in Spanish version? In particular, how large the population is for Spanish-speaking immigrant families with young children, and with limited English proficiency?
Second, the Spanish version of HK is validated using 24-hour dietary recalls, which supposes to be the reference value of dietary intake. However, one major limitation is that dietary intake was assessed only when parents were with their children. What children had at school or being offered by other caregivers is unknown, which is another problem along with the day-to-day variation.
Third, for the reliability of items, why did test-retest reliability not being assessed? Especially when parent-reported dietary intake may subject to recall bias and day-to-day variation, this will be an important data to obtain.
Fourth, there are numerous formatting issues in the manuscript, e.g. inconsistent font size in section 2.6, and the text in strikethrough in Table 3. These are the basic issues that should not appear in high-quality academic manuscripts, bringing concerns on the robustness of study findings.
Nil
Author Response
Author response to Reviewer 3 |
|
First, the novelty of the present study should be elaborated. While HK has already been validated for English-speaking population, how many more beneficiaries will we have after validating HK in Spanish version? In particular, how large the population is for Spanish-speaking immigrant families with young children, and with limited English proficiency?
|
We have added this text to section 1.1: Hispanic/Latinos represent 19% of the United States population and are the ethnic majority in the state of California (40.2%).6 Assessment tools that are tailored for and validated with Hispanic/Latino families with young children are scarce yet necessary to sensibly address nutrition and health inequities affecting a substantial proportion of the population. |
Second, the Spanish version of HK is validated using 24-hour dietary recalls, which supposes to be the reference value of dietary intake. However, one major limitation is that dietary intake was assessed only when parents were with their children. What children had at school or being offered by other caregivers is unknown, which is another problem along with the day-to-day variation. |
In the limitations, we identify the issue you mention. Most of these young children are in Head Start for 2-3 hours, including a lunch. Every parent/child dyad in this study ate a family dinner each night. Parents reported on breakfast, snacks and dinner on the two weekdays. They reported on the 24 hours for the weekend recall. Menus were available for Head Start lunches, but asking parents to guess at the quantity of each food item consumed by their child may introduce substantial systematic bias. |
Third, for the reliability of items, why did test-retest reliability not being assessed? Especially when parent-reported dietary intake may subject to recall bias and day-to-day variation, this will be an important data to obtain.
|
We agree that test-retest reliability is important. Unfortunately, establishing temporal stability was beyond the scope of the study and adding this test could have impacted the precision of our primary estimators. For instance, adding a second administration of Ninos Sanos would add burden to the client and potentially impact the 24-recall data validity. |
Fourth, there are numerous formatting issues in the manuscript, e.g. inconsistent font size in section 2.6, and the text in strikethrough in Table 3. These are the basic issues that should not appear in high-quality academic manuscripts, bringing concerns on the robustness of study findings. |
We apologize. We uploaded our working Word document, as this was allowable per the MDPI author instructions. At MDPI headquarters, it was transferred into a publication format and some new formatting changes occurred. We have made an effort to correct all formatting errors. |

Round 2
Reviewer 2 Report
If the questions to reviewer 3 are properly answered and he/she approves, we consider it acceptable.
I'm sure it's not a problem, but I'm not a native speaker so I can't judge
Reviewer 3 Report
Nil.
Nil.